# Isoflurane anesthesia alters 31P magnetic resonance spectroscopy markers compared to awake mouse brain

Saba Molhemi[1], Rasmus West Knopper[1], Christian Stald Skoven[1],
Thomas Beck Lindhardt[1], Caroline Degel[2], Leif Østergaard[1], Brian Hansen[1]*

**1** Center of Functionally Integrative Neuroscience, Department of Clinical Medicine, Aarhus University, Aarhus, Denmark, **2** Department of Neuroscience, University of Copenhagen, Copenhagen, Denmark

он These authors contributed equally to this work.

* brian@cfin.au.dk

## Abstract

Normal brain function hinges on energy-intensive processes. Consequently, alterations to the brain's metabolic state are common hallmarks in several pathological conditions. Phosphorus Magnetic Resonance Spectroscopy ($^{31}$P MRS) is a non-invasive method for measuring key markers of brain energy metabolism, including adenosine triphosphate (ATP), inorganic phosphate (Pi), and phosphocreatine (PCr), as well as markers for cell membrane phospholipid turnover, phosphomonoester (PME) and phosphodiester (PDE). Preclinical rodent $^{31}$P MRS has so far been done under anesthesia - with isoflurane being one of the most commonly used anesthetic agents. The use of isoflurane in $^{31}$P MRS is a concern, as anesthetics are known to affect neuronal activity and energy metabolism in the brain. Its use therefore comes with a risk of perturbing brain physiology. Awake mouse MRS avoids this and allows the effect of isoflurane to be quantified. Thus, we here compare mouse brain $^{31}$P MRS in awake MR-habituated mice and isoflurane anesthesia. We find that $^{31}$P metabolite levels differ between the awake state and isoflurane anesthesia in mice. Our findings show that low-dose isoflurane anesthesia reduces PCr levels in the mouse brain and is accompanied by decreases in intracellular pH and decreased PME levels.

## 1 Introduction

Phosphorus Magnetic Resonance Spectroscopy ($^{31}$P-MRS) is a non-invasive technique that enables the measurement of metabolites in two major classes within biological tissue: those linked to energy metabolism and those associated with phospholipid membrane turnover. Key energy metabolites include adenosine triphosphate (ATP), phosphocreatine (PCr), and inorganic phosphate (Pi), which are crucial for energy production and consumption [1]. Phosphomonoesters (PMEs) are precursors in the synthesis of membrane phospholipids, while phosphodiesters (PDEs) are the

**Data availability statement:** All data files are available from the dryad database doi:10.5061/dryad.76hdr7t80.

**Funding:** This work was supported by Prof. Leif Østergaard's grant from the Lundbeck Foundation (R310-2018-3455). CD was supported by a grant from Neuroscience Academy Denmark (NAD, grant no. 47178). The funders had no role in study design, data collection and analysis, decision to publish, or preparation of the manuscript.

**Competing interests:** The authors have declared that no competing interests exist.

products of phospholipid breakdown. Together, PMEs and PDEs serve as markers for phospholipid membrane turnover. Furthermore, the chemical shift separation of Pi and $\beta$-ATP relative to PCr provides estimates of intracellular potential of hydrogen (pH) and magnesium ($Mg^{2+}$) concentration [1]. The health and function of all organs in the human body rely on efficient energy substrate utilization for ATP production, phospholipid membrane homeostasis, and general physiological stability. For this reason, $^{31}$P-MRS has become a powerful non-invasive tool for assessing in vivo bioenergetics. Since its first application in humans to study skeletal muscle energy metabolism [2,3], $^{31}$P-MRS has been widely applied to investigate metabolic changes in muscle, heart, liver, and tumors under both physiological and pathological conditions [4–7]. Over the years, $^{31}$P-MRS has proven to be a sensitive modality for detecting alterations in key metabolites such as PCr, ATP, and PME/PDE ratio, which reflect tissue-specific energy demands and dysfunction. More advanced techniques provide insight into energy kinetics by adding magnetization transfer (MT-$^{31}$P-MRS) so that reaction rates of key metabolic enzymes such as creatine kinase and ATP synthase can be quantified [8–14].

In the human brain, energy metabolism predominantly relies on glucose and oxygen utilization to produce ATP through oxidative phosphorylation in mitochondria [15, 16]. A quarter of this energy consumption is dedicated to "housekeeping" processes that maintain cellular structure and integrity, with the maintenance of cell membrane ionic gradients being the most energy-intensive process [15,17]. This is crucial for neuronal signaling, where glutamate is the primary excitatory neurotransmitter, and approximately 80–85% of the brain's total energy consumption is devoted to glutamate-mediated neuronal signaling [15,17,18]. These highly energy demanding processes are reflected in the brain's overall energy consumption. Despite accounting for only ~2% of the body's total weight, the brain accounts for ~20% of the body's total oxygen and glucose consumption [18–21]. At the same time, the concentration of ATP in the brain is low. In vivo human $^{31}$P MRS studies have measured the concentration of ATP in the human brain to be approximately 3 mM [3]. This translates to a small ATP storage of only 2 g in an average-sized human brain of 1.4 kg. MT-$^{31}$P-MRS studies in the human brain in vivo have measured the ATP production rate to be 8.5–9 $\mu$mol g$^{-1}$ min$^{-1}$, which corresponds to the production of ~6 kg of ATP per day [14]. This underscores the extent to which a constant and high ATP production is critical to support the brain's high energy demand. As a result, brain function and health are vulnerable to conditions that disrupt the delivery of substrates for energy production [22], as well as conditions where substrate utilization for ATP production in mitochondria is impaired.

In this context, $^{31}$P-MRS has been used to monitor stroke and traumatic brain injury cases, finding that rapid decreases in ATP levels due to impaired cerebral blood flow (CBF), in severe cases can cause cell damage and death [23]. Moreover, mitochondrial dysfunction is increasingly recognized as a critical factor in various brain disorders, including psychiatric conditions like bipolar disorder and schizophrenia, as well as neurodegenerative diseases such as Alzheimer's and Parkinson's diseases [24–27]. In schizophrenia, $^{31}$P-MRS has revealed decreased ATP levels in the frontal cortex, indicating disrupted energy metabolism [28], while bipolar disorder

shows a reduced PME/PDE ratio, reflecting impaired cellular integrity [29]. $^{31}$P-MRS studies in Alzheimer's disease consistently find altered levels of ATP, PCr, and Pi, which correlate with the severity of cognitive impairment [26]. Parkinson's disease patients also exhibit significant reductions in ATP, PCr and Pi compared to healthy controls [30].

Despite the many applications of $^{31}$P-MRS, its clinical use is not yet common practice. This is mainly due to its low signal-to-noise ratio (SNR) and the need for specialized radio frequency (RF) coils. The $^{31}$P nucleus has a lower gyromagnetic ratio than $^{1}$H, and only 6.7% of the MR sensitivity of $^{1}$H [1]. Although $^{31}$P has 100% natural abundance, the concentration of $^{31}$P metabolites in human tissue is very low [1]. The inherently low SNR of $^{31}$P-MRS results in limited spatial resolution and longer scanning times, making it impractical for routine clinical applications. Consequently, advances in $^{31}$P-MRS are primarily driven by the preclinical research community. High-field scanners ($\geqslant$ 7T) are used to increase SNR and reduce longitudinal relaxation times (T1), giving shorter repetition times (TR) and thus improving SNR per unit time [14,31]. Efforts are also focused on improving RF hardware, as well as refining denoising and post-processing techniques [32,33].

Today, the majority of preclinical research in neuroscience uses rodents, with mice being the primary animal model [34]. It is common practice to perform MRI/S *in vivo* in anesthetized mice to immobilize them [35]. This reduces animal motion and provides stable conditions for averaging to achieve high SNR and images free of motion artifacts. However, research has repeatedly shown that anesthetics affect neuronal activity and brain metabolism [36–40]. Isoflurane, a common general anesthesia used in rodents, is known to reduce cerebral glucose uptake and neuronal firing [38,41,42]. Further, the use of isoflurane requires great care, as it causes respiratory suppression [40,43], decreased arterial blood pressure [44, 45], and widespread vasodilation, which causes loss of functional hyperemia and increased number of capillary stalls [46–48]. More recently, isoflurane has also been found to affect brain microstructure [49] and influence brain solute transport [50]. Collectively, these studies indicate that as MRI and other neuroimaging methods become more sensitive, the presence of isoflurane will have a measurable effect which may cloud data interpretation. Consequently, these effects must be taken into account, or the use of anesthesia avoided. While $^{31}$P-MRS in awake mice has been demonstrated [51] (see also Discussion) further work is needed to quantify the effects of isoflurane on $^{31}$P spectra and to assess whether spectra acquired in awake animals can achieve comparable quality and reliability to those obtained under conventional conditions using isoflurane anesthesia. This necessitates experiments directly comparing measurements obtained in the awake state to those obtained under isoflurane anesthesia, assessing the spectral quality, and reproducibility of awake $^{31}$P-MRS relative to the established isoflurane protocol.

In light of these considerations, the aim of our study is two-fold. Firstly, we perform single-voxel $^{31}$P MRS in mice in both the awake state and under isoflurane anesthesia, demonstrating the feasibility of routine acquisition of awake $^{31}$P-MRS data. This is achieved using an in-house Mouse Cradle Suspension System (MCSS) designed for use in both awake and/or anesthetized conditions [52]. Secondly, we perform quantitative comparison of $^{31}$P-MRS data from awake and isoflurane anesthetized mice (males and females) to investigate the effects of isoflurane on $^{31}$P metabolite levels and ratios. Our study ushers in the use of awake mouse $^{31}$P-MRS for investigation of brain metabolism and cell phospholipid membrane turnover under physiological conditions not affected by anesthesia. This is valuable for gaining a deeper understanding of brain energy supply, consumption and failure.

## 2 Materials and methods

### 2.1 Animals and housing

For repeatability tests 26-week-old C57BL/6JRj male (n = 3, body weight = 25.3 $\pm$ 1.16 g) mice were used. Our comparisons of isoflurane anesthetized and awake $^{31}$P-MRS scans used 21-week-old C57BL/6JRj male (n = 4, body weight = 23.9 $\pm$ 2.22 g) and female (n = 4, body weight = 20.3 $\pm$ 0.83 g) mice. Upon arrival at our animal facility, the mice were allowed 2 weeks of acclimatization. Throughout, the mice were housed in ventilated cages with a 12-h light-dark cycle in a controlled vivarium with 50% humidity and room temperature at 23 °C. Water and food were provided *ad libitum*.

All experimental procedures were approved by the Danish Animal Experiments Inspectorate (permit number: 2019-15-0201-00285). One male and one female mouse had their head plates gnawed excessively, leading to their exclusion in the awake and isoflurane comparison experiment. At the end of the experiments, mice were deeply anesthetized with isoflurane and euthanized by an intraperitoneal overdose of pentobarbital, in accordance with our animal experiment permit.

## 2.2 Surgical procedures

After the acclimatization period, all mice had a head plate surgically implanted. We use an in-house head plate design, addressing some limitations of the previous head collars used in our lab [52,53]. The horseshoe-shaped head collar is bulky, restricts movement, and causes abrasions on the mouse's shoulders over time. Moreover, by design, the head collar holds the mouse's head in place by fixing the collar to the bed's collar stage. This occupies the space where the front legs are usually placed and results in an unnatural posture for the mouse. While habituation to this setup is well established, in our experience, the initial response to awake MR habituation, compared to awake optical imaging, shows higher stress levels in the mice. The new head plate stabilizes the mouse's head from the sides, allowing space for normal posture similar to awake optical habituation setups. Due to its minimalistic design, the head plate does not restrict the mouse's movements when caged, nor is interaction with cage enrichment interfered with. MR-compatible head plates for awake MRI/S scans must be non-magnetic and are primarily made of 3D-printed plastic parts. These parts are eventually gnawed when mice, especially females, are not single-housed. If not mitigated, this gnawing destroys the head plate and limits the feasibility of longitudinal studies involving repeated measurements. To address this, the head plate is used together with a 3D-printed plastic cover that can be easily clamped on when the animal is caged and removed for experiments. In this manner, any gnawing damage affects the interchangeable plastic cover rather than the head plate itself (see Fig 1g–1i).

The surgical procedure began with the mice being briefly anesthetized using 4% isoflurane mixed in a 1:1 ratio of medical air (0.4 mg/L) and oxygen (0.4 mg/L). The mice were then positioned on the stereotaxic frame using the tooth bar for anesthesia and two non-rupture earbars to fixate the head. A heating pad and rectal probe maintained the core body temperature at $37 \pm 0.5$ °C. During the whole procedure the isoflurane was regulated in the ranges of 0.9–1.4% to avoid excessive respiratory suppression while maintaining sufficient anesthetic depth. The mice further received a subcutaneous injection of lidocaine in the scalp region. Carprofen, ampicillin, and buprenorphine were administered intraperitoneally at least 30 min. prior to the first incision. The head fur was removed using a trimmer and hair removal gel (Pure hair removal cream, Veet), then cleaned with 70% ethanol. A small amount of skin was then removed to expose the skull (see Fig 1a). The skull was cleaned, scored, and prepared with a thick layer of superglue (Power gel, Loctite) applied at the occipital lobe and along the interfrontal fissure (see Fig 1b). Before implantation, a ceramic rod was placed with super glue superficially in the groove of the head plate, aligned so that the rod was a seamless continuation of the superficial surface of the head plate (see Fig 1c). The headplate was then mounted on the skull above the area with superglue. The tip of the 3D-printed MR head plate was positioned approximately 1 mm above lambda (see Fig 1d). Next, a small amount of super glue accelerator (Insta-set Accelerator, Easyshoe) was applied while gently pressing the tips of the ceramic rod and the 3D printed MR head plate to the skull. A second thin layer of superglue was then applied to the skull without exceeding the thickness of the ceramic rod (see Fig 1e and 1g). A final thin layer of dental cement (Meliodent Rapid Repair Powder, Kulzer GmbH, Hanau, Germany mixed with Meliodent Rapid Repair Liquid 0.5L, Forstec Dental AB, Malmö, Sweden), was applied, and the mouse was moved to a preheated recovery chamber (see Fig 1f). Post-surgery, the mice received antibiotics and pain relief subcutaneously with carprofen, ampicillin, and buprenorphine for at least the following three days.

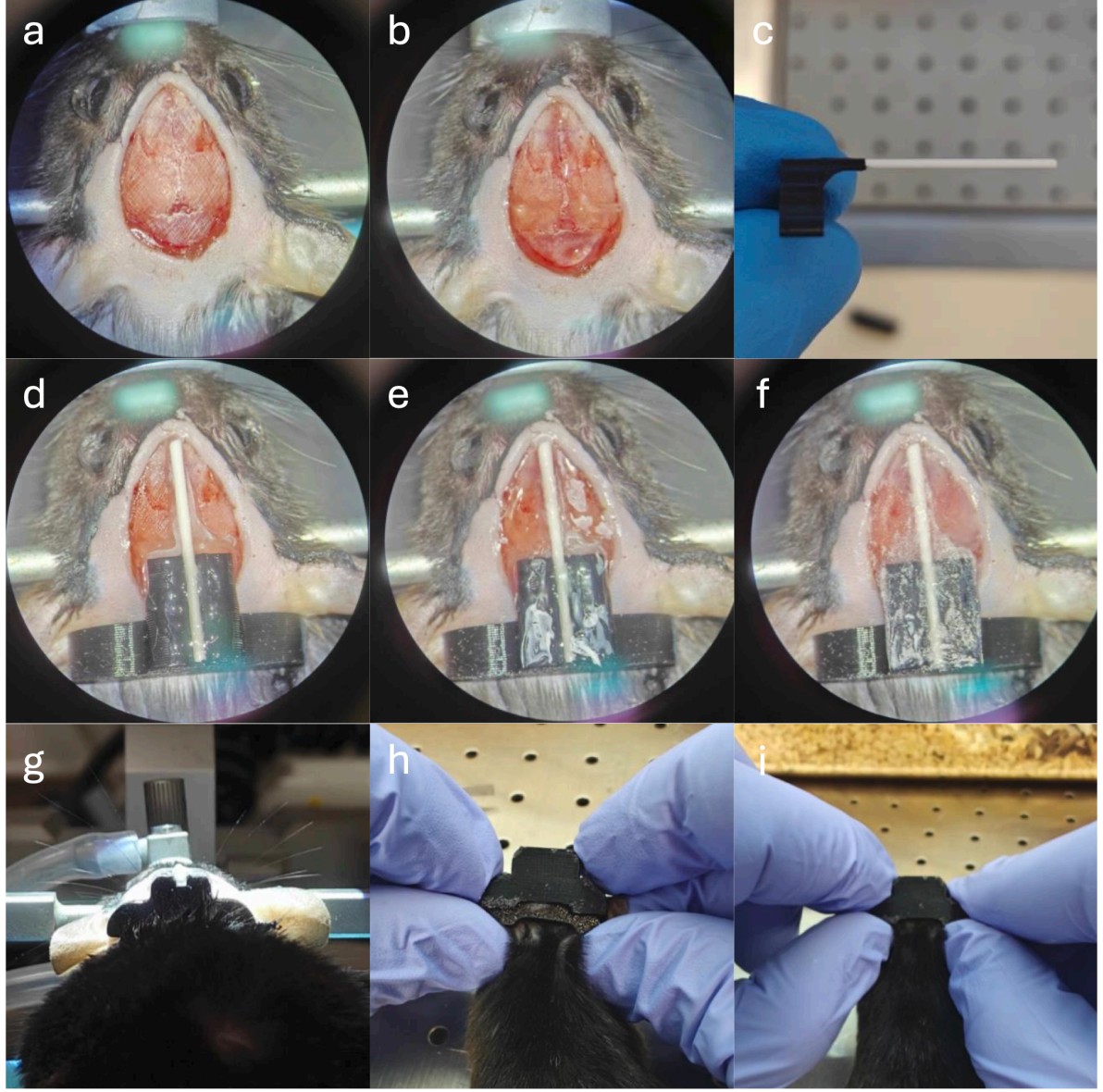

**Fig 1**. **MR head plate surgery.** Surgical procedure for implantation of an MR-compatible head plate for awake mouse MR scanning. The stl files for the head plate and cover plate are found in the supplementary S1 and S2 Files.

## 2.3 MRS habituation

When performing awake MR scans in mice, the mice must be habituated to head and body restraint and MRI noise, to avoid high stress and arousal states. Habituation to these conditions ensures that the collected data are closer to normal arousal states at rest. For this, we used our previously established habituation protocol [53]. This involved gradually exposing the mice to the MR environment by head fixation for 20 minutes on the first day. On subsequent days this duration increased by 20 minutes per day. On the fourth day, MR sounds were added to the habituation environment. On day 9, the mice had reached the maximum duration of 180 minutes, which was maintained until day 14. However, based on previous work [53] we used an updated version of our habituation box, allowing six mice to be trained in parallel while

still being visually isolated from each other. For this, a new habituation cradle has been added and features a socket into which the new head plate fits snugly. The cradle also includes vertical tracks along which a mock surface coil is lowered until it contacts the head. The coil is then fixed in position using side screws that lock the head plate to the habituation cradle (see Fig 2a–2c).

## 2.4 Preparations for awake and anesthetized MRS

For these studies we used an in-house coil mouse cradle suspension system (MCSS) compatible with custom mouse cradles suitable for both awake and anesthetized $^{31}$P-MRS. This setup is described elsewhere [52] including detailed build and operation instructions. After completing the habituation protocol, mice were ready for awake $^{31}$P-MRS scans. The mice were initially anesthetized in a small chamber with a mixture of 4% isoflurane, 1:1 ratio of medical air (0.4 mg/L) and oxygen (0.4 mg/L), and then moved to the MRS cradle. The mice were placed on the MRS cradle and covered with a piece of tissue and tape, similar to the habituation cradle setup, while maintaining isoflurane anesthesia at approximately 1% isoflurane with 0.4 mg/L medical air and 0.4 mg/L $O_2$. A cut 3 mL single-use pipette was used as a nose cone to guide isoflurane from the MR bed's anesthesia tube to the mouse (see Fig 3a). A respiration pillow sensor (RS-301, Small Animal Instruments, Inc.) for monitoring respiration rate (RR) was placed on the MR cradle beneath the mouse's chest. Next, the MRS cradle was slid forward so that the mouse's head was positioned below the dual-tuned $^{31}$P/$^1$H surface coil, with

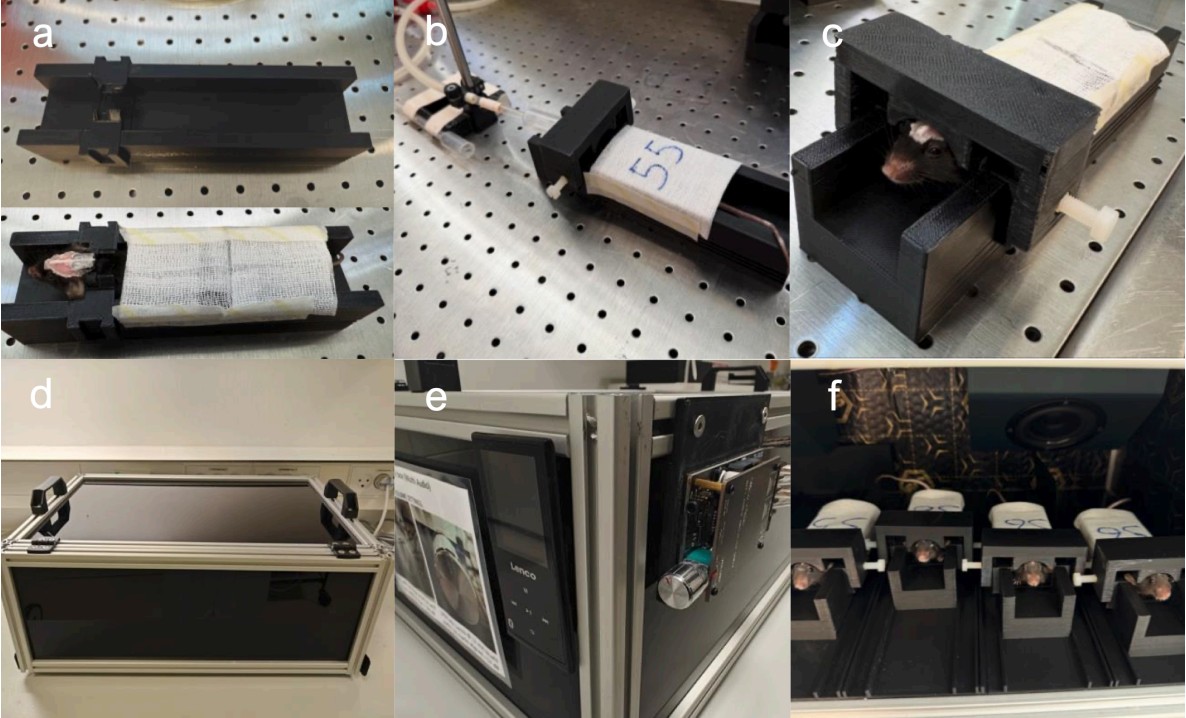

**Fig 2. Setup for awake MRS habituation.** (a) A 3D-printed cradle with and without a mouse placed in the cradle. (b and c) The mouse, lightly anesthetized with 1% isoflurane, is secured to the cradle, and a mock surface coil is lowered along the cradle's vertical rails until it touches the mouse's head and is then locked in place by tightening the screws to the cradle. (d) The habituation cradle with the lid closed. (e) The speakers (SPX-30M 3", Monacor) are connected on the back of the box to an Bluetooth amplifier board (ZK-1001B Mono 100W), and paired via Bluetooth to a media player (Lenco XEMIO-760 BT, XEMIO760BTSW, Lenco) for MR scan sound playback. (f) The interior of the habituation box, can accommodate up to six cradles, shown with the large interior speakers in view along with four mice next to each other. The stl files for the habituation bed, head plate holder and mock surface coil are available in the supplementary S3, S4 and S5 Files.

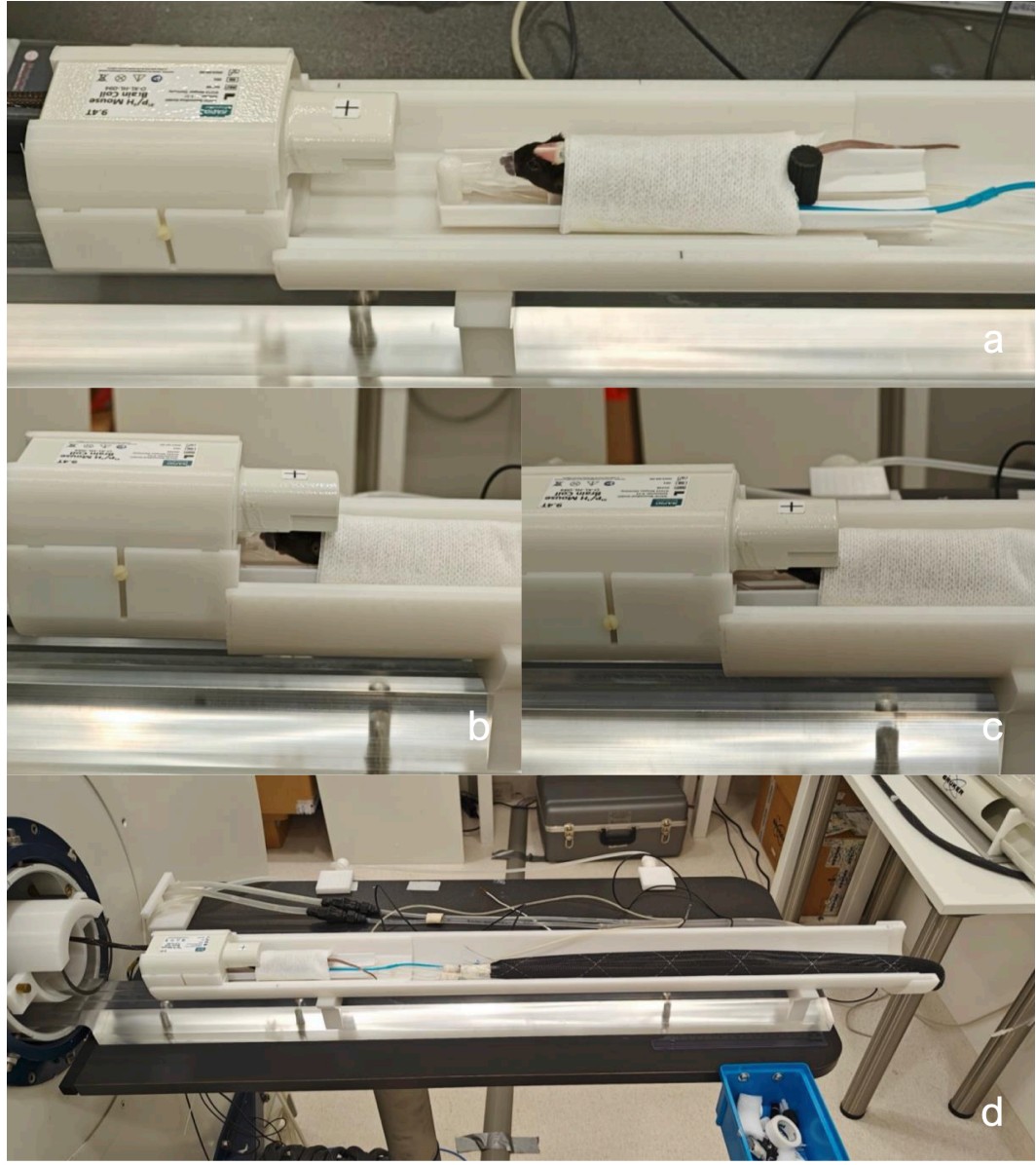

**Fig 3. MRS setup.** Setup for performing awake or anesthetized $^{31}$P-MRS From [52]. Reprinted with permission. The stl file for the mouse cradle is available in the supplementary S6 File.

the rear tips of the ears aligned with the coil's isocenter (see Fig 3a–3b). The surface coil was lowered onto the mouse's head until it physically touched, then secured with side screws (see Fig 3b–3c). Anesthesia was turned off, allowing the mice to wake up. The pipette tube was still positioned to the mouse, fitting closely around its mouth and nose. To ensure sufficient air during awake scans, the mouse was provided with 0.4 mg/L medical air. The entire setup was inserted into the scanner ensuring positioning of the mouse's head in the scanner isocenter. The MRS cradle was connected to water inflow and outflow tubes which were connected to a water heating circulator (SC 100, Thermo Scientific). During anesthetized scans the water heating circulator was turned on to ensure that the mouse temperature was maintained in the

range of 37 $\pm$ 0.5 °C. In awake scans this was not needed because the awake animal can regulate its body temperature naturally.

## 2.5 Single voxel $^{31}$P MRS Acquistion

All data were acquired on a horizontal 9.4 T preclinical MRI system (Biospec 94/20, Bruker, Ettlingen, Germany) with BGA12S gradients and a dual-tuned $^{1}$H/$^{31}$P mouse brain surface coil (O-XL-HL-094-01843, Rapid, Germany). This coil has loop diameters of 15 mm ($^{31}$P) and 21 mm ($^{1}$H). All $^{31}$P data sets were acquired using an Image Selected In Vivo Spectroscopy (ISIS) pulse sequence [54]. As preparation for ISIS scans optimal inversion and 90-degree flip angle pulses were identified. For this, we performed a series of $^{31}$P single pulses with incremental increases in power (W) on a phantom solution containing 3 mM ATP in saline (Thermo Scientific, adenosine 5'-triphosphate, Catalog no. R1441). The $^{31}$P reference power was determined from the $^{31}$P single pulse scan that produced the highest signal. Additionally, the $^{1}$H reference power was acquired through reference power calibration using a 1 mm thick coronal slice through the phantom solution. These reference powers were then used to calculate the $^{1}$H/$^{31}$P ratio. Each *in vivo* data collection started with a $^{1}$H localizer scan followed by a B0 map with a field of view (FOV) of 32 mm ×32 mm ×32 mm and image size of 128 ×128 ×128 with 3 averages. The quality of the localized shim was assessed by the full width at half maximum (FWHM) of the water peak obtained from a 6 mm ×3.7 mm ×6 mm voxel carefully placed in the mouse brain to avoid signal from non-brain tissue (see Fig 4). The voxel geometry (size, position) from the localized shim were transferred to the ISIS sequence, and the $^{1}$H reference power was calibrated using a 1 mm thick coronal slice positioned inside the ISIS voxel and aligned with its superior boundary. The optimal $^{31}$P power for the ISIS was derived from the $^{1}$H reference acquired by the coronal slice and the $^{1}$H/$^{31}$P ratio. The ISIS pulse sequence was acquired with the following parameters: a TR of 5000 ms, 90 ISIS averages, 6009 Hz acquisition bandwidth, and 10600 acquisition points. Three 180-degree inversion pulses (2.058 ms, 16000 Hz) and a 90-degree excitation pulse (0.05 ms, 25600 Hz) were used. Due to its reliance on combining signals from eight scan cycles, ISIS is more prone to motion-induced artifacts and frequency drift compared to single-shot localization methods. Thus, a new B0 map and localized shim was always obtained prior to each ISIS acquisition.

## 2.6 Repeatability test

The reliability of our $^{31}$P-MRS acquisition protocol and hardware setup was first evaluated by repeatability tests. One aim of this study is to assess the reliability of awake $^{31}$P-MRS by comparing it to scans performed under isoflurane anesthesia. This is done by performing sequential awake followed by anesthetized ISIS scan acquisitions. To assess potential

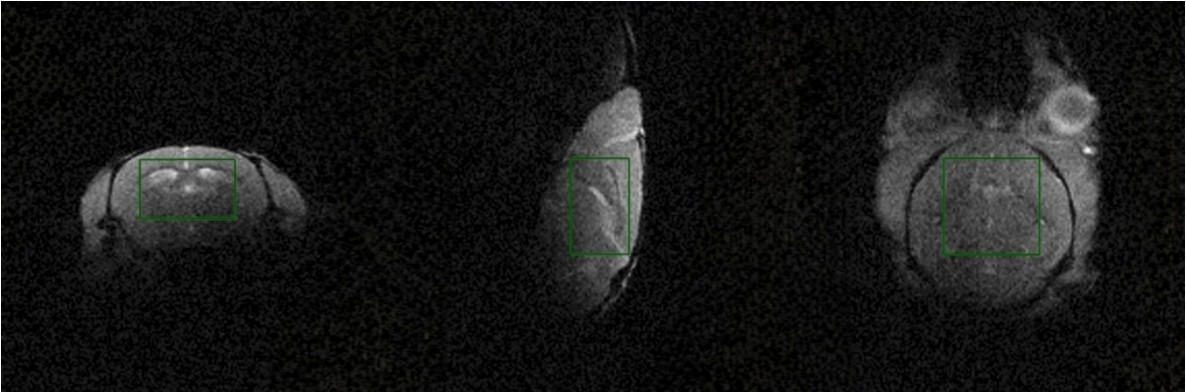

**Fig 4**. **Whole-brain $^{31}$P MRS.** Illustration of the positioning of a 6 mm x 3.7 mm x 6 mm voxel for whole-brain $^{31}$P MRS acquisition using ISIS.

intra-mouse variability between two consecutive acquisitions, and independently of the awake versus isoflurane comparison, back-to-back ISIS scans were performed in three C57BL/6JRj male mice under isoflurane anesthesia. Each ISIS scan lasted 1h, consistent with similar single-voxel $^{31}$P MRS studies in the mouse brain [33,55,56]. Due to animal welfare, our experimental license restricts awake MRI scans to a maximum duration of 2.5 h. The B0 map takes 15 minutes and is performed once for each ISIS, which limits the possibility of multiple repeated awake scans. However, these consecutive anesthetized scans allowed evaluation of short-term measurement stability within a single session, at standard isoflurane anesthesia conditions used in preclinical MRI [57].

For future awake $^{31}$P-MRS studies allowing comparison between treatment, phenotype, or other groups, assessing inter-mouse and inter-session variability under awake conditions is essential. To this end, three mice were scanned once daily over three consecutive days. This design allowed evaluation of variability arising from mouse placement, hardware setup, positioning, and inherent biological differences. The coefficient of variation (CoV) for each mouse was calculated as $100 \times$(standard deviation / mean AUC) and expressed as a percentage, for the most robust metabolites PCr, $\gamma$-ATP and their ratios PCr/$\gamma$-ATP. The reported CoV represents the mean across all three mice.

## 2.7 Awake and anesthetized ISIS scans

The first ISIS acquisition was performed on awake mice. After discontinuing isoflurane administration, the mice were allowed 10 minutes of recovery before initiating the 15-minute B0 mapping. This gave the mice a total of 30 minutes from cessation of anesthesia to the start of the first ISIS scan. Following the first ISIS scan, the mice were anesthetized with isoflurane while in the MRI scanner bore (induction at 4 %). This order was chosen to avoid data contamination from post-anesthesia effects, which are known to linger for a considerable time (>1h) even after short exposure duration (0.5–1h) [49,58,59]. A period of 2–3 minutes was allowed to ensure the animal was anesthetized. Then, isoflurane was lowered and B0 mapping was repeated so that a new localized shim could be obtained. Following this, the anesthetized ISIS acquisition was performed. During this scan the RR was kept between 80–120 breaths per minute by adjusting the isoflurane (maintained in the range of 0.5%–1.2%) mixed with a 1:1 ratio of medical air (0.4 mg/L) and oxygen (0.4 mg/L), while keeping the body temperature at $37 \pm 0.5$ °C as described. A fume extractor was positioned at the front-end of the MCSS acrylic tube to reduce isoflurane accumulating in the bore.

## 2.8 Spectral processing

Spectral data were all processed in TopSpin (version 4.4.0) using the same processing steps and values. This involved apodization with a line broadening factor of 12 Hz, followed by Fourier transformation. Manual phase correction was centered on PCr and PCr set to 0 ppm as is standard [60]. Phase correction parameters were kept consistent between the two consecutive anesthetized scans in the intra-variability test, as well as between the awake and anesthetized spectra for each mouse. An automatic baseline correction with a 5th-order polynomial (ABSG = 5), was found to produce the best achievable baseline correction, and was applied to all spectra. The FWHM and SNR of the PCr peak was determined using the 'calculate peak width' and the SiNo functions.

## 2.9 Data analysis

The processed spectra were exported to MATLAB (Mathworks, Natick, USA) for further processing and analysis using in-house script. The area under the curve (AUC) of each metabolite was calculated by integrating the signal intensity within the predefined chemical shift ranges (in ppm) specific to each metabolite: 7.5 to 5.8 for PME, 5.6 to 4.1 for Pi, 3.7 to 2.5 for PDE, 0.5 to –0.5 for PCr, –2.1 to –3.2 for $\gamma$-ATP, –7 to –8 for $\alpha$-ATP, and –15.7 to –17 for $\beta$-ATP. Noise estimation was based on two spectral regions without metabolite signal peaks: 13 to 8.5 ppm and –18 to –23 ppm. The mean signal from these regions was subtracted from each data point to perform baseline correction. The mean and standard deviation of the noise was then calculated, and a threshold was defined as the average noise level plus one standard deviation.

Only values exceeding this threshold within the defined metabolite ranges were included in the integration (see S1 Fig). Further, ratios of PCr/$\gamma$-ATP and PCr/Pi were calculated using the AUCs. Within the ppm range for Pi, the chemical shift at the local peak was determined. Intracellular pH was then determined using the Henderson-Hasselbalch equation [61]:

$$pH_i = 6.77 + \log_{10}\left(\frac{\delta_{Pi} - 3.29}{5.68 - \delta_{Pi}}\right) \tag{1}$$

In this equation, (pH$_i$) represents the intracellular pH, $\delta_{pi}$ is the difference in chemical shift between Pi and PCr. The constants 6.77, 3.29, and 5.68 are derived from calibration data [61]. The values 3.29, and 5.68 are the acid ($H_2PO_4^-$) and alkaline ($H_2PO_4^{2-}$) end points, respectively, for a Mg concentration of 1.36 mM corresponding to brain cytosol [62–66]. The constant 6.77 is the acid dissociation constant (pKa) for Pi.

Respiration and core body temperature were recorded using an MR-compatible Small Animal Monitoring and Gating System (model 1030; SAII) in combination with the PC-SAM software (version 7.07; SAII). Due to occasional suboptimal contact between the respiration pad and the mouse chest, a custom MATLAB script was developed to process the respiration data. This consisted of a global thresholding by calculating the mean and standard deviation of the respiration rate within the scan window. Any values outside the $\pm 1$ standard deviation from the mean are temporarily marked as outliers. Next, a 10-second moving average window was applied to the respiration signal to smooth high-frequency fluctuations (see S2 Fig). For statistical analysis, paired Student's t-tests were performed on the normalized AUCs, metabolite ratios, and intracellular pH.

## 3 Results

The first aim of this study was to evaluate whether $^{31}$P-MRS can be performed in the awake mouse brain with reliability comparable to that achieved under anesthesia. For this, we first assessed the shim performance by comparing the FWHM of the $^1$H peak signal in the awake and isoflurane anesthetized state for all animals. No significant difference in FWHM was observed between the first and second ISIS acquisitions in the intra-variability test, indicating stable shim performance across back-to-back repeated shim performance. However, the FWHM in awake mice was $30.3 \pm 2.8$ Hz, which was significantly ($p = 0.004$) higher than the FWHM during anesthesia, at $26.8 \pm 3.1$ Hz (see Table 1). Likewise, the FWHM across all awake inter-variability tests was significantly ($p = 0.01$) larger ($27.2 \pm 2.3$ Hz) than in the anesthetized condition during the intra-variability tests ($19.9 \pm 0.9$ Hz). Although higher, the shim quality remained sufficient to avoid any significant differences in spectral resolution quality, as indicated by the SNR of PCr, $\gamma$-ATP and Pi both between intra- and inter-variability tests and between awake and anesthetized scans. In addition, no significant differences were found between female and male mice in neither awake nor anesthetized states.

**Table 1. Summary of experimental metrics reported as mean $\pm$ standard deviation. Asterisks (*) and daggers (†) indicate pairs of values that differ significantly ($p < 0.05$). Shim FWHM was significantly lower in awake conditions compared to isoflurane-anesthetized conditions.**

| Experiment | Shim FWHM (Hz) | PCr SNR | $\gamma$-ATP SNR | Pi SNR |
|---|---|---|---|---|
| Intra-variability 1. ISIS | 20.03 $\pm$ 0.64 | 22.56 $\pm$ 1.33 | 6.52 $\pm$ 1.00 | 4.49 $\pm$ 0.25 |
| Intra-variability 2. ISIS | 19.70 $\pm$ 1.21 | 19.11 $\pm$ 1.70 | 6.28 $\pm$ 1.41 | 4.42 $\pm$ 1.23 |
| Intra-variability all ISIS | †19.87 $\pm$ 0.88 | 20.83 $\pm$ 2.33 | 6.40 $\pm$ 1.10 | 4.46 $\pm$ 0.80 |
| Inter-variability 1. ISIS | 28.27 $\pm$ 0.64 | 19.73 $\pm$ 4.52 | 5.97 $\pm$ 1.67 | 4.61 $\pm$ 1.52 |
| Inter-variability 2. ISIS | 27.20 $\pm$ 2.23 | 20.42 $\pm$ 2.16 | 6.01 $\pm$ 0.84 | 4.70 $\pm$ 0.07 |
| Inter-variability 3. ISIS | 26.10 $\pm$ 3.37 | 21.39 $\pm$ 5.18 | 5.70 $\pm$ 1.33 | 4.50 $\pm$ 0.86 |
| Inter-variability all ISIS | †27.19 $\pm$ 2.25 | 20.52 $\pm$ 3.68 | 5.89 $\pm$ 1.16 | 4.60 $\pm$ 0.88 |
| Awake | *30.25 $\pm$ 2.82 | 22.75 $\pm$ 2.81 | 6.63 $\pm$ 1.21 | 4.22 $\pm$ 0.70 |
| Anesthetized | *26.83 $\pm$ 3.11 | 20.25 $\pm$ 3.31 | 6.36 $\pm$ 1.07 | 4.31 $\pm$ 1.39 |

Results from our inter-variability tests are shown in Fig 5 which shows AUCs and ratios for PCr, γ-ATP, Pi, and their ratios from three mice each scanned once per day for three days. A high consistency is seen between mice and across scan days. While some variation is seen no systematic trends are observed. Importantly, the PCR/γ-ATP-ratio displays high stability across mice and scan days (Fig 5, top right panel). The CoV within mice was PCr: 7.83 ± 3.41%, γ-ATP: 9.55 ± 4.66% and PCr/γ-ATP: 6.88 ± 0.93%. Across-mice CoV was PCr: 11.92%, γ-ATP: 12.29% PCr/γ-ATP: 4.23%.

Fig 6 details the intra-variability results. Here, the repeated measurements of the key metabolites PCr and γ-ATP showed good repeatability. PCr/γ-ATP performed well with little variation. Although here the second ISIS scan is seen to consistently yield higher PCr/γ-ATP ratios. Considering the stability observed in our inter-variability tests we believe this is due to physiological variations between the first and second ISIS scan. The respiratory rate (RR) was consistently ~20 breaths per minute lower in the second ISIS compared to the first ISIS during the intra-variability experiments. It was not feasible to maintain the same respiration rate (RR) between the two ISIS scans, despite reducing isoflurane below 0.5% while ensuring adequate ventilation during the second ISIS. Pi exhibited a larger variability between the repeated measures. To evaluate awake RR for signs of animal stress we used an MR-compatible respiration pad for RR monitoring. While sub-optimal for awake mouse RR measurements, five of the nine awake ISIS scans from the inter-variability test yielded artifact free RR readouts with RR values in the range of 103–148 breaths per minute (see S8 Fig).

## 3.1 Awake vs anesthetized

Next, we examine whether metabolite levels differ between the awake and isoflurane anesthetized states (see Fig 7). We see no difference between male ($n = 3$) and female ($n = 3$) mice in this study. Therefore, sex is ignored in the following and data from males and females are pooled (effective $n = 6$). Here, the PCr level was significantly ($p = 0.002$) reduced in

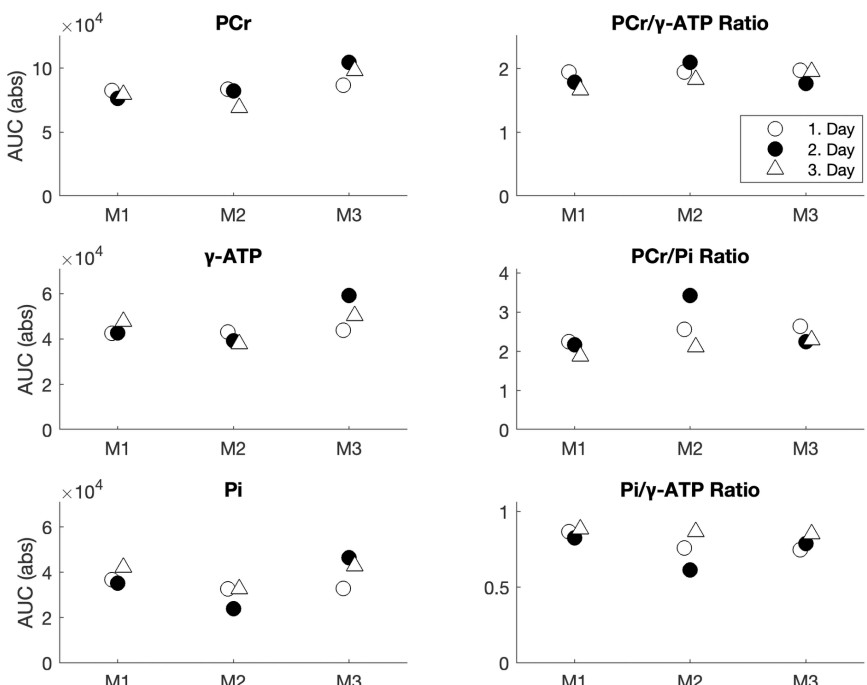

**Fig 5. Inter-variability during awake.** Metabolite levels and ratios are acquired from three mice over three consecutive days.

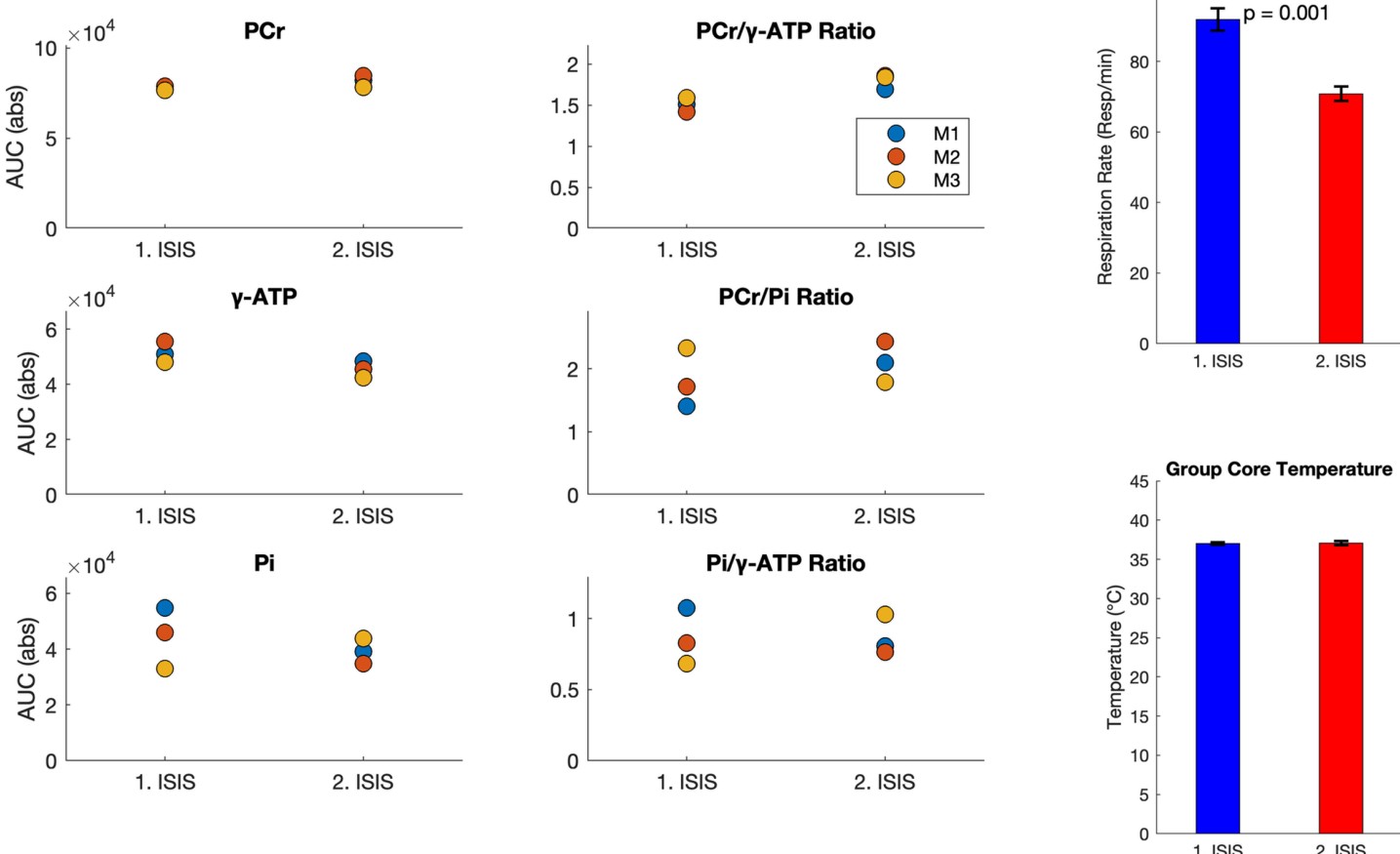

**Fig 6**. **Intra-variability under isoflurane anesthesia.** Metabolite levels of Pi, PCr, $\gamma$-ATP and their derived ratios from two consecutive ISIS acquisitions in isoflurane anesthetized mice (n = 3). Core body temperature and respiratory rate are shown between scans. See S4 and S5 Figs for all metabolite levels and ratios.

isoflurane anesthetized mice, with a 12.9% reduction compared to their levels in the awake state (see Fig 8). The PCr/$\gamma$-ATP, PCr/Pi and Pi/$\gamma$-ATP ratios are indicators of cellular energy status and phosphorylation potential [67] , with reductions in PCr/Pi as an indicator of metabolic crisis [2,68]. However, no significant differences in the PCr/$\gamma$-ATP, PCr/Pi or Pi/$\gamma$-ATP ratio were found between awake and anesthetized states (see S7 Fig). Likewise, no significant differences in the PME/PDE ratio, the marker of phospholipid membrane turnover, were found (see S7 Fig). Lastly, we did observe a significant ($p = 0.02$) reduction in intracellular pH in the anesthetized state compared to the awake. In the awake brain a pH of $7.13 \pm 0.03$ was found, whereas under isoflurane this value is shifted down to $7.05 \pm 0.06$ (see Fig 9).

## 4 Discussion

Awake mouse $^{31}$P-MRS was previously reported alongside other data examples in [51] as part of a demonstration of a cradle system suitable for awake mouse MRI. Still, more in-depth characterization of awake $^{31}$P MRS is needed to understand the effects of isoflurane anesthesia and to guide in choice of methods and study design. For this reason, we provide detailed analysis of awake mouse $^{31}$P readouts obtained with a openly accessible hardware setup [52] compatible with commercially available volume and surface RF-coils. Our study compares the awake state to the isoflurane anesthetized

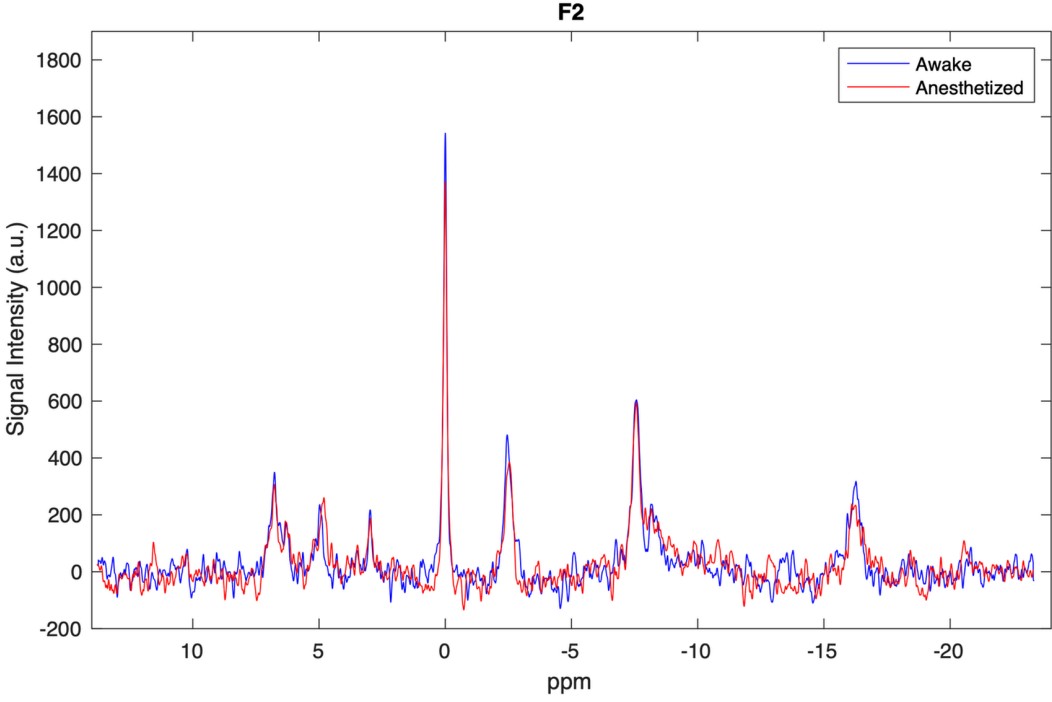

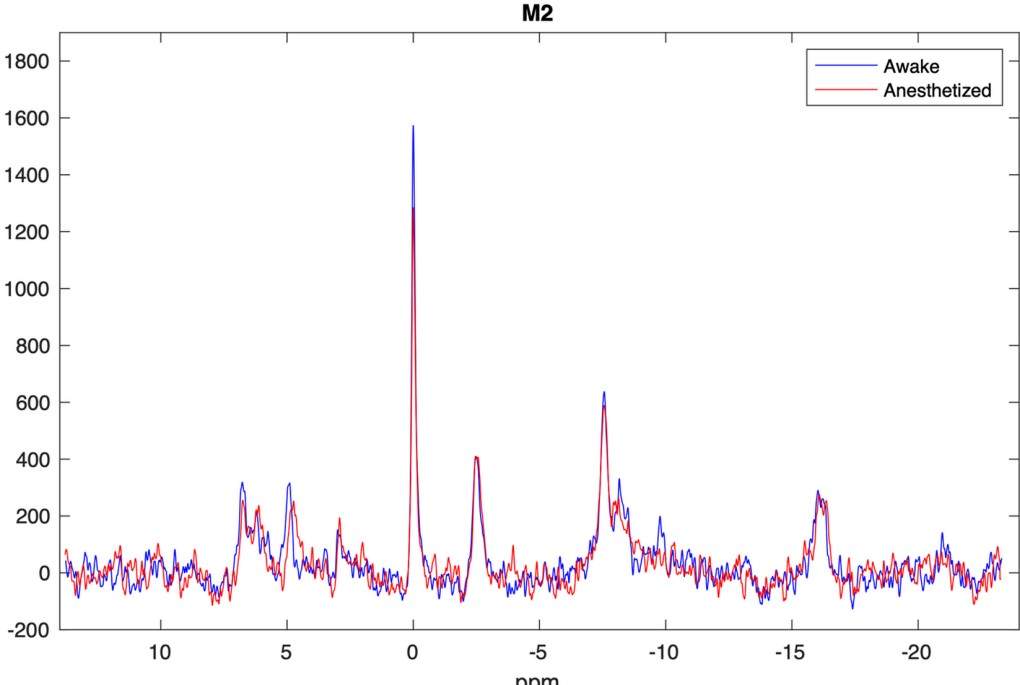

**Fig 7. Spectra assessment.** The $^{31}$P spectra for a female (F2) and male (M2) mouse in both awake and anesthetized states. For an overview of each individual mouse during awake and anesthetized scans see S6 Fig)

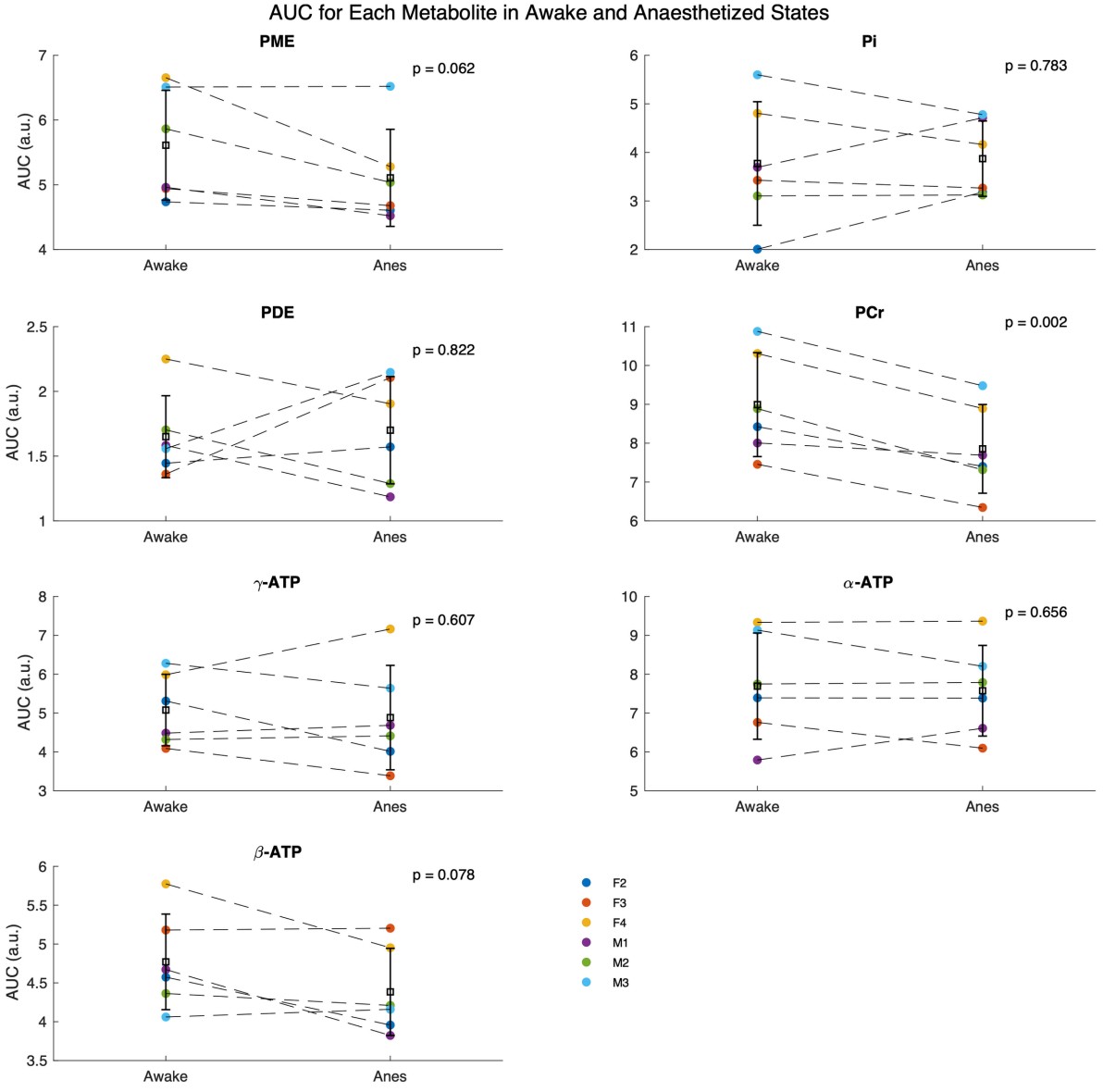

**Fig 8. $^{31}$P metabolite levels.** The Raw AUC for PME, Pi, PDE, PCr, $\gamma$-ATP, $\alpha$-ATP and $\beta$-ATP for each mouse in each condition with dashed lines.

state using both male and female mice. We demonstrate the feasibility of routine (1.5 h scan protocol) $^{31}$P-MRS in awake, MR-habituated mice, providing characterization of shim quality and $^{31}$P spectral assessment.

Although better shim was achieved under isoflurane anesthesia (a result likely to be found for any anesthetic), the localized shim performed well in the awake state too, providing reliable $^{31}$P spectra with high quality and SNR. The study by Hyen-Man et al. [33] also performed mouse whole-brain ISIS $^{31}$P-MRS at 9.4T (voxel size = 6.0 mm ×4.0 mm ×6.0 mm, 128 ISIS averages) and employing advanced low-rank denoising achieved a PCr SNR of 15.6 $\pm$ 1.7 while ours was $\geq$ 19.1. Regarding the repeatability tests, Pi, PME and PDE generally exhibited larger variability between the repeated measures. This is to be expected, due to their smaller peak size, where even minor fluctuations have a relatively larger effect on the measured AUC [69,70]. This increased variability was also reflected in the ratios PCr/Pi and Pi/$\gamma$-ATP, which

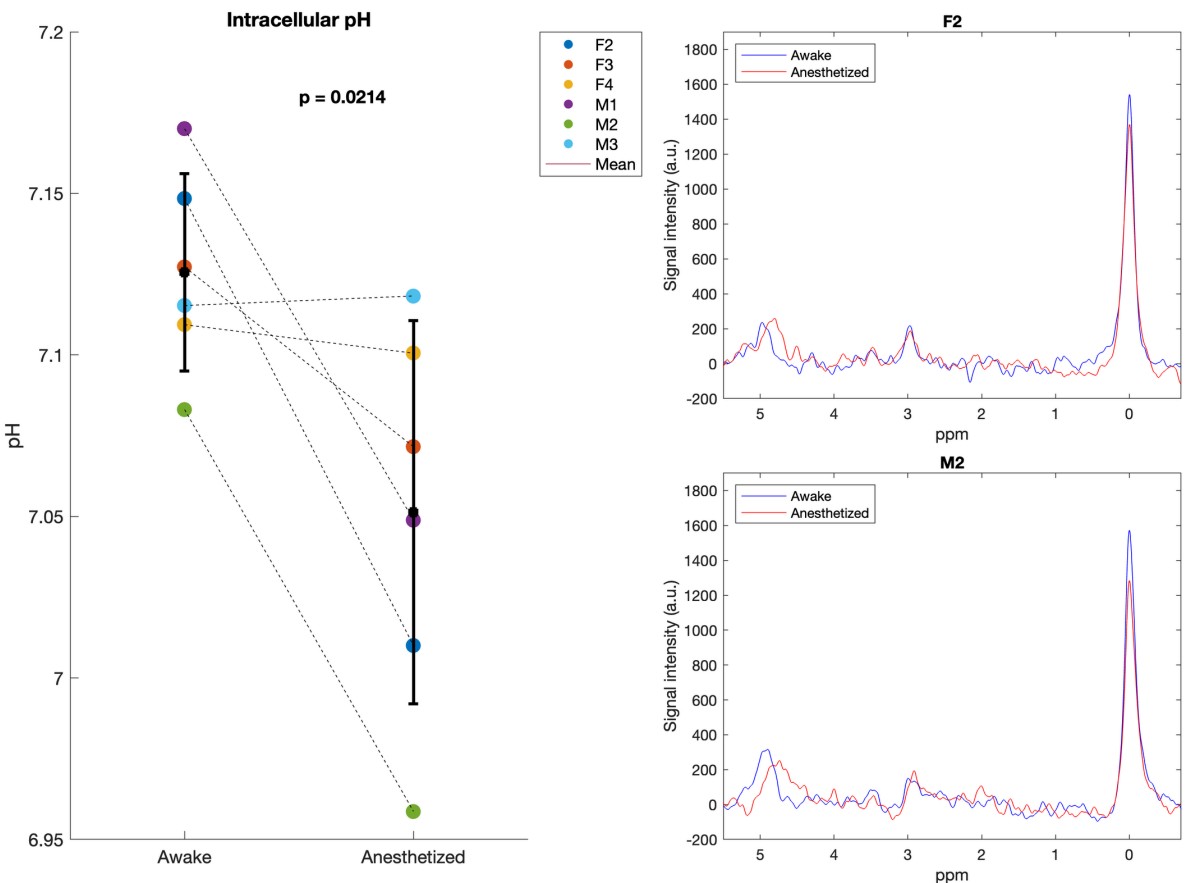

**Fig 9**. **Intracellular pH levels.** The intracellular pH in the anesthetized state is significantly lower compared to the pH in the awake state ($n = 6$). To the right, the chemical shift between Pi and PCr in both states for the female mouse F2 and the male mouse M2 is shown (see S3 Fig for the chemical shift between awake and anesthetized states for each mouse).

showed more variance compared to the PCr/$\gamma$-ATP ratio. This altogether demonstrates that awake $^{31}$P-MRS is feasible with spectral quality comparable to that obtained under isoflurane anesthesia. This allows future study of brain energy metabolism without the perturbing influence of anesthetics on brain physiology. Specifically, isoflurane anesthesia is known to affect brain energy consumption [39,71], and blood flow [44,45], as well as microstructure [49]. Isoflurane's widespread vasodilatory effect has also been reported to increase the number of stalled capillaries, compromising micro-circulation [46–48]. However, the impact of isoflurane on $^{31}$P metabolites, compared to measurements under awake, normal physiological conditions, has not yet been assessed. Our study therefore also aimed to investigate whether low-dose isoflurane anesthesia, as commonly used in preclinical rodent imaging, has detectable effects on $^{31}$P-MRS data. To this end, we employed a low-dose isoflurane protocol ($0.5 − 1.2\%$) maintaining RR at $91 \pm 5.1$ per minute with a core body temperate of $37 \pm 0.5°C$. We found PCr to be significantly reduced under isoflurane compared to the awake state. Furthermore, we also observe an unavoidable drift towards slower respiration rates during longer periods of isoflurane anesthesia as e.g. seen when performing repeated ISIS acquisitions for our intra-variability tests. Here, the second ISIS scan was found to consistently yield higher PCr/$\gamma$-ATP ratios than the first ISIS scan (see Fig 6) despite good repeata-bility observed in our inter-variability scans (see Fig 5). Surprisingly, this increase seems to be driven by a drop in $\gamma$-ATP

as PCr is seen to be stable (see Fig 6). This is surprising as one would expect PCr to decrease before a decrease in $\gamma$-ATP would manifest. While the shift in PCr/$\gamma$-ATP ratio might therefore be due to data variability the fact that this happens alongside a decrease in respiration rate between the first and second ISIS acquisition (see Fig 6) is a concern. Next, we offer a biological interpretation of this observation and its implications for MRI/S experiments using isoflurane anesthesia.

*Biological interpretation*

We found PCr to be significantly reduced under isoflurane compared to the awake state. Previous studies using [1]H MRS also found isoflurane to reduce PCr levels [72]. This is expected, as PCr acts as an energy buffer, helping maintain a high ATP/ADP/AMP ratio to avoid catabolic processes during decreased levels of ATP, ADP, and AMP [73,74]. Therefore, during conditions where energy consumption exceeds the aerobic ATP production rate from ATP synthase (ADP + Pi + $2H^+_{out} \rightarrow$ ATP + $2H^+_{in}$), PCr maintains ATP levels through an increased reverse reaction flux of creatine kinase (PCr + ADP $\rightarrow$ ATP + Cr) [75,76]. The reduction in PCr in isoflurane anesthetized mice therefore suggests insufficient oxidative ATP production. It is well-established that isoflurane causes dose-dependent reduction in cerebral energy metabolism, along with respiratory suppression [77,78]. Previous work has also shown that lactate accumulation increases as respiratory depression becomes more pronounced, and is proportional to isoflurane concentration [40,43]. Therefore, it is no surprise that insufficient oxidative ATP production might occur during isoflurane anesthesia. However, we saw no significant differences in the PCr/$\gamma$-ATP and PCr/Pi ratios between the awake/isoflurane states, indicating that no severe alterations to energy metabolism occured in our low-dose isoflurane protocol.

Nevertheless, the observed decrease in PCr was accompanied by a decrease in intracellular pH. This could potentially be due to an increase in lactate concentration, which, as previously mentioned, is known to accompany respiratory suppression [40,43]. The pK constant in the Henderson-Hasselbalch equation is temperature-dependent. One might therefore speculate whether the shift in pH could be due to a potential shift in animal temperature between the awake and anesthetized state. We do not believe this to be the case, as animal temperature was closely monitored and controlled in the range of $37 \pm 0.5$ °C during our experiments. This was possible because our MRS cradle has an integrated water container, connected to a heating water circulator. Here, only animal core temperature was monitored. However, studies have shown that brain temperature does not match core temperature in both humans and animals [79]. Mouse brain cortex temperature is typically slightly lower than core temperature [80] and is actively regulated [81,82], but affected by even brief isoflurane anesthesia [80]. Since we have no way of monitoring brain temperature during scans this might be a potential source of variation in our results. However, the small variations in core temperature will likely result in similar small fluctuations in brain temperature especially since our experiments are done in mice with intact skulls in contrast to open skull preparations where differences between core and brain temperature can be substantial if not actively corrected [82]. For these reasons, we believe that the measured core temperature is a good proxy for brain temperature. Minor variations will have little effect on our intracellular pH estimates because near 37°C a one degree increase in temperature decreases pH by only 0.01 to 0.02 [83,84]. In our hands, maintaining a RR of 80–120 breaths per minute required early reduction of the isoflurane concentration from 1.2% to below 1%, and in some cases closer to 0.5%. Pilot experiments showed that if isoflurane was not adjusted in this way, the RR dropped below 90, approaching 60–70 breaths per minute. Therefore, deliberate regulation of isoflurane-% is necessary to maintain the RR in the ranges of 80–120 breaths per minute. Our study showed that even when the RR is maintained at $91 \pm 5.1$ per minute a slight reduction in PCr is seen compared to awake, which indicates early signs of insufficient aerobic ATP production. This production deficiency is likely proportional to the degree of respiratory suppression induced by isoflurane, and to the associated resulting increase of lactate [40,43].

In case of ATP depletion the $Na^+/K^+$ pump halts, causing an increased intracellular $Na^+$ and increased water influx [85–87]. This results in cell swelling and decreased extracellular space which is a hallmark of cytotoxic edema [85–87]. In such cases the $Na^+/H^+$ gradient can be disrupted, resulting in intracellular $H^+$ accumulation [88], an additional source for reduced intracellular pH due to isoflurane. This is unlikely the case in our study, given the level of ATP was unchanged between the two states. However, conditions with decreased ATP levels could arise in isoflurane anesthesia protocols

with RR < 90, which future studies should investigate. The reduced PCr indicates that even under ideal conditions, isoflurane can lower pump efficiency with cell swelling as a near-unavoidable consequence. The average RR in awake resting mice is 165 breaths per minute in C57BL/6J [89]. However the normal RR range during wakefulness for mice is 100–180 breaths per minute and a reduction down to 50 % of this is considered acceptable (50–90 breaths per minute) in preclinical MRS [35]. The RR range used in this study aligns with this and the Animal Care Committee Guidelines from the University of British Columbia, which recommend maintaining it at 80–120 breaths per minute [90]. However, our study suggests that the RR range at which $^{31}$P metabolite concentrations are unaffected is above 90 breaths per minute for both sexes of adult C57BL/6JRj mice. Further, our data indicate that experiments under isoflurane should be completed in less than 1.5hr unless blood gasses are monitored or the animal ventilated.

Other $^{31}$P-MRS methods may be able to shed light on the mechanisms underlying our findings. MT-$^{31}$P-MRS is a promising utilization of $^{31}$P-MRS as it allows for direct and accurate measurement of ATP and PCr production rate, contrary to other MR-based alternatives more widely used today. These MRI techniques allow measurements of the cerebral metabolic rate of $O_2$ ($CMRO_2$) or glucose ($CMRO_{glu}$), and are proxy markers for the brain ATP production rate. As MT-$^{31}$P-MRS measures the PCr and ATP production rate, its measurements are expected to be proportionally affected by reductions in neuronal activity and glucose consumption associated with isoflurane anesthesia [38,41,42]. While comparative awake and anesthetized mouse MT-$^{31}$P-MRS experiments would further our understanding of anesthetic agents' effects on the ATP and PCr production rate, these experiments are lengthy and challenges the feasibility for prolonged (>1.5 h) awake experiments.

*Methodological considerations*

Our study also has direct implications for conventional $^{31}$P brain MRS. Firstly, our results show a detectable difference in $^{31}$P metabolite levels between the awake state and isoflurane anesthesia. This must be taken into account when using $^{31}$P-MRS to study aspects of brain energy metabolism in isoflurane anesthetized rodents. Otherwise there is a risk of faulty data interpretation or actual effects being shadowed by perturbations originating from isoflurane. Furthermore, MRI/S is increasingly used in combination with e.g. optical *in vivo* methods such two-photon microscopy [91] where awake scans are the norm. In such cases, we recommend that *awake* MRI/S should be performed, as the brain otherwise is not in the same physiological state during scans - hampering direct data comparison.

Our habituation procedure is a refinement of the method presented in [53] using a modified head plate and a high capacity habituation box. We observed no visible stress indicators (e.g. distress squeak, ocular foam) even in initial training stages suggesting a low baseline stress level likely achieved by the use of the new head plate and careful handling of animals in the days leading up to habituation training. Similarly, after habituation RR rates (103–148 breaths per minute) fall within the expected range for awake, non-stressed mice reported in C57BL/6J [89] (see S8 Fig). While our study demonstrates that awake $^{31}$P-MRS is feasible, it is not always a practical option. In certain acute animal models—such as traumatic brain injury or stroke—anesthetic protocols are mandatory to prevent discomfort. Moreover, current awake mouse MRS methods require surgery, post-surgical recovery, extensive handling, and MR habituation training. These steps are not only time-consuming but also stressful for the animals, and they are incompatible with longitudinal studies in moderate to severe brain injury models [92]. In experimental cases where anesthesia is unavoidable, a combination of dexmedetomidine/medetomidine and isoflurane is a viable option in rodents [93–95]. In functional MRI, blood oxygen level dependent (BOLD) experiments rely on intact neurovascular coupling dynamics, which are severely affected by the widespread vasodilation caused by isoflurane alone. Co-administration with dexmedetomidine/medetomidine allows for a lower isoflurane dose, thereby mitigating this effect [58,93,95]. Moreover, dexmedetomidine/medetomidine is an $\alpha$-2 adrenergic agonist, mimicking noradrenaline's vasoconstrictive mechanism [96] and counteracting the loss of muscle tone and thus the respiratory suppression induced by isoflurane. However, in any anesthetized experiment, special care must be taken when choosing anesthetics, as they are not freely interchangeable [97]. Different anesthetics cause distinct physiological perturbations affecting the nervous, cardiovascular, respiratory, and other systems in unique ways [98–101]. The choice of anesthetic should be guided by the specific research question with comparisons made to control animals

undergoing the same anesthetic regime. Careful characterization of an anesthetic's effects on the relevant study readouts is therefore essential, as one anesthetic may be well suited for certain studies yet unsuited for others. Evidently, alternatives are needed as isoflurane is now known to perturb the brain to a degree detectable with perfusion MRI [102], contrast based MRI [103], functional MRI [58,95], diffusion MRI [49] and 31P-MRS. Further work is needed to determine whether other types of anesthesia cause fewer physiological perturbations, making them more suitable for MRI/S. The MCSS used here is designed to investigate such cases by allowing seamless transition between awake and anesthetized 31P- and 1H-MRS, using any types of gas or infusion anesthesia.

## 5 Conclusion

We show that awake mouse 31P-MRS is feasible and demonstrate that 31P metabolite levels are affected by isoflurane anesthesia - even at the low-dose commonly used in other rodent MRI/S studies. This effect must be considered when interpreting 31P-MRS and MRI data from anesthetized animals.

## Supporting information

**S1 Fig. 31P Spectra processing.** This figure illustrates the set ranges for each individual metabolite and the threshold (green dashed line) used calculate the AUC for each metabolite.
(TIFF)

**S2 Fig. Raw and processed biometry data from M3 from the first ISIS during the intra-variability test.** Temperature measurements are unprocessed and include only measurements during ISIS acquisition, while respiratory data are processed.
(TIFF)

**S3 Fig. Chemical shift of Pi.**
(TIFF)

**S4 Fig. 31P Intra-variability.**
(TIFF)

**S5 Fig. 31P Intra-variability - ratios.**
(TIFF)

**S6 Fig. 31P Spectra.** Each individual 31P Spectrum from male and female mice in both awake and anesthetized states.
(TIFF)

**S7 Fig. Awake and anesthetized metabolite ratios.** The AUC where used to determine the PCr/$\gamma$-ATP and PCr/Pi ratios. No significant changes where found when comparing awake and anesthetized states.
(TIFF)

**S8 Fig. Awake respiration rate.**
(TIFF)

**S1 File. MR Head Plate.** STL file of the custom-designed MR head plate. Recommended to be 3D printed with a high layer height, slow print speed, and slightly increased filament temperature and extrusion multiplier (at 105–110%) to ensure optimal strength.
(STL)

**S2 File. Cover Plate.** STL file of the cover plate used in conjunction with the MR head plate. Designed to protect the head plate from gnawing.
(STL)

**S3 File. MR Habituation Bed.** STL file of the MR habituation bed used for acclimate mice to the MR environment. Designed for use in training protocols prior to awake scanning.
(STL)

**S4 File. Head Plate Holder.** STL file of the head plate holder for the habituation sessions. Allows the head plate to positioned in the socket, securing it in place.
(STL)

**S5 File. Mock Habituation Surface Coil.** STL file of the mock surface coil used during habituation sessions. Mimics the geometry of the actual surface coil to allow for realistic training without active hardware.
(STL)

**S6 File. Mouse Cradle – top part.** STL file of the updated mouse cradle top part, replacing the version described in [52]. Assembly follows the same instructions.
(STL)

## Acknowledgments

We are grateful for assistance from Claire Wary, Bruker Biospin.

## Author contributions

**Conceptualization:** Saba Molhemi, Brian Hansen.

**Data curation:** Saba Molhemi, Rasmus West Knopper, Christian Stald Skoven.

**Formal analysis:** Saba Molhemi.

**Funding acquisition:** Leif Østergaard.

**Investigation:** Saba Molhemi, Christian Stald Skoven, Thomas Beck Lindhardt, Caroline Degel.

**Methodology:** Saba Molhemi, Brian Hansen.

**Project administration:** Saba Molhemi, Brian Hansen.

**Resources:** Brian Hansen.

**Supervision:** Brian Hansen.

**Validation:** Saba Molhemi.

**Visualization:** Saba Molhemi.

**Writing – original draft:** Saba Molhemi.

**Writing – review & editing:** Saba Molhemi, Christian Stald Skoven, Brian Hansen.

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
