## [Decision Letter · Decision Letter 0]

6 Aug 2025

PONE-D-25-37692Isoflurane anesthesia alters 31P Magnetic Resonance Spectroscopy markers compared to awake mouse brainPLOS ONE

Dear Dr. Molhemi,

Thank you for submitting your manuscript to PLOS ONE. After careful consideration, we feel that it has merit but does not fully meet PLOS ONE’s publication criteria as it currently stands. Therefore, we invite you to submit a revised version of the manuscript that addresses the points raised during the review process.

We look forward to receiving your revised manuscript.

Kind regards,

Jochen Leupold

Academic Editor

PLOS ONE

Journal Requirements:

2. To comply with PLOS One submissions requirements, in your Methods section, please provide additional information regarding the experiments involving animals and ensure you have included details on (1) methods of sacrifice, (2) methods of anesthesia and/or analgesia, and (3) efforts to alleviate suffering."

This work was supported by Prof.\ Leif Østergaard’s grant from the Lundbeck Foundation (R310-2018-3455). CD was supported by a grant from Neuroscience Academy Denmark (NAD, grant no.\ 47178).

5. Please amend your list of authors on the manuscript to ensure that each author is linked to an affiliation. Authors’ affiliations should reflect the institution where the work was done (if authors moved subsequently, you can also list the new affiliation stating “current affiliation:….” as necessary).

6. Please ensure that you refer to Figure 5 and 7 in your text as, if accepted, production will need this reference to link the reader to the figure.

7. We are unable to open your Supporting Information file [Cover plate.stl, Head plate holder for habituation.stl, Mock habituation surface coil.stl, Mouse cradle top part.stl, MR – Habituation bed.stl, MR head plate.stl]. Please kindly revise as necessary and re-upload.

Additional Editor Comments :

Dear Authors,

your manuscript was reviewed by two experts with many years of experience in preclinical MRI/MRS. Both of them appreciate your work but they suggest a number of modifications and clarifications. In particular, both reviewers express the lack of focus of the discussion part.

Please address these issues in a revised version of the manuscript. Thank you.

Minor comment from the editor: Ref. 8 and 9 is doubled in the list of references.

Kind regards,

Jochen Leupold

Reviewers' comments:

Reviewer's Responses to Questions

**Comments to the Author**

1. Is the manuscript technically sound, and do the data support the conclusions?

Reviewer #1: Yes

Reviewer #2: Partly

2. Has the statistical analysis been performed appropriately and rigorously?

Reviewer #1: Yes

Reviewer #2: Yes

3. Have the authors made all data underlying the findings in their manuscript fully available?

Reviewer #1: Yes

Reviewer #2: Yes

4. Is the manuscript presented in an intelligible fashion and written in standard English?

Reviewer #1: Yes

Reviewer #2: Yes

5. Review Comments to the Author

Reviewer #1: The manuscript “Isoflurane anesthesia alters 31P Magnetic Resonance Spectroscopy markers compared to awake mouse brain” presents a technically sound study comparing brain energy metabolism in awake versus isoflurane-anesthetized mice using 31P-MRS. The authors demonstrate that even low-dose isoflurane significantly reduces phosphocreatine (PCr) levels and intracellular pH, without affecting ATP concentrations or PCr/ATP ratios. The study is strengthened by meticulous attention to experimental detail, e.g. in the design of the custom head-fixation and cradle system, animal habituation protocol, 31P power calibration and, in particular, the careful control of physiological parameters. These efforts convincingly establish the feasibility of awake 31P-MRS and provide a reference point for future studies on brain metabolism under physiological versus anesthetized conditions. While the methodology is robust, several aspects — such as repeatability under awake conditions and the interpretation of specific metabolic shifts and the extensive biological interpretations in the discussion would strongly benefit from a larger number of animals and/or experiments.

However, some minor issues and questions remain:

Many of the Figure and suppl. Figure references as well as the references to suppl. Stl files are wrong. I have marked the ones I saw, but please revise the manuscript carefully.

Animal prep, Page 4, line 140:

Please clarify the timing of the carprofen, Buprenorphine and ampicillin. In the text it read like it was injected i.p. right before incision. However at least for Buprenorphine and ampicillin recommendations are 30-60min prior to surgery. Also most of the drugs are recommended s.c. Why did you decide for i.p.?

Caption Fig 1: “… supplementary files 4 4.” The Manuscript has stl files called, e.g., ‘MR head plate.stl’

please fix reference.

Page 5, Caption Fig 2: again the reference to supplementary files “4, 4.” please fix.

Page 6, Caption Fig 3: again the reference to supplementary files “4.” please fix.

Line 219: see Fig 4 should be Fig 2?

Page 7, line 239ff: with an ISIS scan time of 60min, why was it impossible to do two awake acquisitions back to back? The duration of “awake MRI” probably starts the moment the mice are put into the head mount. However, thanks to the careful prior calibration of the 1H and 31P Reference power, the setup time (localizers) shouldn’t be that long?

Page 8, line 287: ref to Suppl. Fig 4 is wrong,. supplied SFig 4 is biometric results. Please fix.

line 297ff: It seems the respiration pad was not only used for online monitoring but was recorded. By which system? Please add in methods.

line 300ff: Was this respiration processing performed on both awake and anesthetized? Which respiration

rates (awake or anesthetized) are shown in Sfig 4 (see also Caption on page 14) ? Since its a

specific animal, which one (M1..3, F1..3) ?

Page 9, line 319: again a wrong reference to suppl. Fig 4. please fix.

Table 1: Please try to add a visual indication which values are compared for the significant t-test results for better understanding, e.g. mark both compared values by the same marker (its only two pairs).

Line 320, 324: Fig X → Fig 5? Please fix

Line 333ff: These are the RR under anesthesia. What were the RR in awake state? Did those also change over time (habituation, increasing discomfort/panic, …) ?

Page 10, Caption fig 9: wrong reference to Sfig 4, please fix.

Line 352: References to Fig 9a and 9b – but figure doesn’t have a and b label. Please fix.

Chapter discussion:

There are almost two pages of discussion of the results and their biological interpretation, containing elaborate arguments, thought chains and connections with other known facts, which is appreciated. But it also sometime seems a bit of overinterpreting the results of a lower PCr content in awake mice with an n=3, while ATP and ratios were non significant. I do appreciate tying the PCr change and the pH change together with other prior knowledge, but maybe there is potential to shorten this and tighten the arguments?

Line 368: please fix typo texttimes

Page 12, line 452, 460: Is RR in line 452 a general value? Or measured in your experiment?

please add the measured RR values from awake state as comparison for the discussion about RRs in anesthesia.

Page 13, line 473ff: the authors suggest awake mouse MT-31P-MRS should facilitate measuring the true, physiological ATP creation rates without isoflurane influence. This seems possible, but a few problems might be addressed as well, like limited scantime and MT-31P-MRS beeing notoriously long scans. Motion stability for longer scantimes in ISIS might still become a problem, even if its not actual head motion, but e.g. body motion during breathing or by fighting against the restraint, which could causie B0 fluctuations in the head. Please address these issues as well.

Line 479: Here authors claim, ATP suppression by anesthesia is substantial – however, that is not based on shown data nor is it backed by a reference. Please add support for that claim.

Line 483 needs a closing paranthesis after the figure 7 reference.

Line 482ff: While the argument, that basic 31P MRS analysis was already able to show significant PCr decrease, a more fancy analysis might have been able to find more accurate ATP changes and PCr/ATP ratios as well. However, the small n=3 would still weaken those results, so maybe indeed leave it at this.

Section “Methodological considerations”

In fMRI in rats and mice isoflurane does indeed vasodilate the brain vessels and leaves no no capacity to further open up the vessels for a increased blood flow in BOLD experiments, effectively canceling BOLD signal for isoflurane at 2%. However, a combination anesthesia of Medetomidine together with (very) low levels of isoflurane recover the BOLD responsiveness of the rodent brain. In comparison to your (low) isoflurane protocol, added Medetomidine might allow to lower isoflurane even further, maybe keeping the respiration rate >100, possibly also reducing the ATP suppression. Please add a discussion of regarding this well described anesthesia combination.

Page 14: author contributions

Line 523: … surgeries, SM, SM and RWK performed ..... : the second SM is not required/wrong, please fix

Support information:

The naming of the S1 File, S2 file is not the same as on the downloadable files, please fix

References:

Some references contain broken html formatting codes (e.g. 21 !sup?14 with ! And ? Inverted), please fix.

Also Ref 29, 71, 72, 88

Reviewer #2: The authors have performed 31P MRS in awake and isoflurane anesthetized mice. They use a mouse cradle suspension system they have previously developed and compare repeated measurements as well as awake versus anesthetize male and female mice. The sample sizes (number of animals) are rather low. However, for the purpose of this study it appears acceptable to present significant values based on such low sample numbers. The study is experimentally sound and has been performed according to the state-of-the-art. The authors state two aims of this study. First, they want to show that 31P MRS can be performed in awake mice. Second, they want to assess the effects of isoflurane anesthesia on 31P MRS by comparing awake versus anesthetized mice. Regarding the first aim, the study provides convincing evidence that 31P MRS in awake mice is feasible. Regarding the second aim, the relevance of the findings does not become fully clear and several aspects in the discussion lack the necessary depth.

Therefore, the following points should be addressed by the authors:

Major points:

1. The study compares awake mice with isoflurane anesthetized mice. Presentation of the results and its discussion creates the impression that these data allow for general statements about awake versus anesthetized investigations. However, only one defined anesthetic condition was used. While it may represent a widely used regimen, other studies may use different isoflurane concentrations, and more importantly other anesthetics. Especially medetomidine is becoming more and more popular. It should become clear from the paper that the investigations do not allow for general statements beyond the anesthetic regimen applied.

2. The discussion is very broad and lacks focus. It touches upon many aspects of physiological and metabolic differences between awake and anesthetized states and cites a large number of references. However, several aspects are only discussed superficially, not with the required depth. Some are only loosely linked to the presented investigations and do not support the conclusions. Of course, it is the authors choice what they want to discuss. However, given the already large number of citations, I would recommend to focus the discussion on the relevant points, in order to avoid the paper (and refence list) to become exceedingly lengthy. Some examples:

a. One example is the discussion about brain clearance. This topic, if not omitted, requires much more in-depth discussion with a much wider scope of citations to address the topic adequately. (remark in this context: I do not see how ref. 44 fits on page 3).

b. Another example is cell swelling. If not omitted, it should be mentioned that altered diffusivity has been measured in dependence on brain activity (see DOI: 10.1371/journal.pbio.2001494 , 10.1126/science.1241224, 10.1186/s12987-023-00443-2 )

c. Effects of other anesthetic drugs are not discussed in sufficient detail, but seem important for the conclusions.

d. PCr changes under isoflurane have previously been observed by 1H MRS (https://doi.org/10.1016/j.neuroimage.2012.12.020) which might be worth discussing.

3. The authors state that body temperature of the mice was kept constant. However, there are reports that brain temperature can differ substantially from the core temperature. This seems not unlikely in cases where the skull is exposed. Do the authors have any possibility to measure (or estimate) the actual brain temperature? Otherwise, no statements about brain temperature should be made and a potential variability should be discussed as a possible source of error.

4. The authors state that breathing rate varied between scans, but do not discuss the implications in sufficient detail. It has been shown previously that CO2 level depends on breathing rate, which in turn may have impact on pH. More discussion is needed how this may influence the obtained results.

Minor points:

1. Which reference line was used for the 31P ppm scale?

2. Ref 52 requires updating

6. PLOS authors have the option to publish the peer review history of their article (what does this mean?). If published, this will include your full peer review and any attached files.

Reviewer #1: No

Reviewer #2: **Yes: **Cornelius Faber

---

## [Author Response · Author response to Decision Letter 1]

4 Sep 2025

Our response to the points:

1.

Done.

2.

Done

3. and 4.

This has been addressed in the comment section.

5.

Done.

6.

Done.

7.

This has been adressed in the comment section.

8.

Done.

---

## [Decision Letter · Decision Letter 1]

16 Sep 2025

Isoflurane anesthesia alters 31P magnetic resonance spectroscopy markers compared to awake mouse brain

PONE-D-25-37692R1

Dear Dr. Molhemi,

We’re pleased to inform you that your manuscript has been judged scientifically suitable for publication and will be formally accepted for publication once it meets all outstanding technical requirements.

Kind regards,

Jochen Leupold

Academic Editor

PLOS ONE

Additional Editor Comments (optional):

Reviewer #1:

Reviewer #2:

Reviewers' comments:

Reviewer's Responses to Questions

**Comments to the Author**

1. If the authors have adequately addressed your comments raised in a previous round of review and you feel that this manuscript is now acceptable for publication, you may indicate that here to bypass the “Comments to the Author” section, enter your conflict of interest statement in the “Confidential to Editor” section, and submit your "Accept" recommendation.

Reviewer #1: All comments have been addressed

Reviewer #2: All comments have been addressed

2. Is the manuscript technically sound, and do the data support the conclusions?

Reviewer #1: Yes

Reviewer #2: Yes

3. Has the statistical analysis been performed appropriately and rigorously?

Reviewer #1: Yes

Reviewer #2: Yes

4. Have the authors made all data underlying the findings in their manuscript fully available?

Reviewer #1: Yes

Reviewer #2: Yes

5. Is the manuscript presented in an intelligible fashion and written in standard English?

Reviewer #1: Yes

Reviewer #2: Yes

6. Review Comments to the Author

Reviewer #1: All comments have been addressed by the authors in the revised manuscript. I recommend publication of the revised version.

Reviewer #2: The authors have addressed all my previous comments. I have no further points concerning content of the manuscript. However, the sequence of uploaded figures (and supplements) seems to be mixed up and should be corrected before publication.

7. PLOS authors have the option to publish the peer review history of their article (what does this mean?). If published, this will include your full peer review and any attached files.

Reviewer #1: **Yes: **Karl-Heinz Herrmann

Reviewer #2: **Yes: **Cornelius Faber, University of Münster, Germany

---

## [Editor Report · Acceptance letter]

PONE-D-25-37692R1

PLOS ONE

Dear Dr. Molhemi,

I'm pleased to inform you that your manuscript has been deemed suitable for publication in PLOS ONE. Congratulations! Your manuscript is now being handed over to our production team.

Kind regards,

on behalf of

Dr. Jochen Leupold

Academic Editor

PLOS ONE